# A Comparative Analysis of Neurosymbolic Methods for Link Prediction

**Guillaume Olivier Delplanque**                    GUILLAUME.DELPLANQUE@INRIA.FR
**Luisa Sophie Werner**                             LUISA.WERNER@INRIA.FR
**Nabil Layaïda**                                   NABIL.LAYAIDA@INRIA.FR
**Pierre Genevès**                                  PIERRE.GENEVES@INRIA.FR
*Tyrex team, Univ. Grenoble Alpes, CNRS, Inria, Grenoble INP, LIG, 38000 Grenoble, France*

**Editors:** Leilani H. Gilpin, Eleonora Giunchiglia, Pascal Hitzler, and Emile van Krieken

## Abstract

Link prediction on knowledge graphs is relevant to various applications, such as recommendation systems, question answering, and entity search. This task has been approached from different perspectives: symbolic methods leverage rule-based reasoning but struggle with scalability and noise, while knowledge graph embeddings (KGE) represent entities and relations in a continuous space, enabling scalability but often neglecting logical constraints from ontologies. Recently, neurosymbolic approaches have emerged to bridge this gap by integrating embedding-based learning with symbolic reasoning. This paper provides a structured review of state-of-the-art neurosymbolic methods for link prediction. Beyond a qualitative analysis, a key contribution of this work is a comprehensive experimental benchmarking, where we systematically compare these methods on the same datasets using the same metrics. This unified experimental setup allows for a fair assessment of their strengths and limitations, bringing elements of answers to following key questions: How accurate are these methods? How scalable are they? How beneficial are they for different levels of provided knowledge and to which extent are they robust to incorrect knowledge?

## 1. Introduction

Knowledge Graphs (KGs) are structured representations of entities, their attributes, and relationships, widely used across domains to organize and interconnect vast amounts of data. A KG is composed of factual triples in the form $(head, relation, tail)$. For example, the triple $(Paris, CapitalOf, France)$ represents the fact that Paris is the capital of France. KGs can be enriched with an ontology, which defines rules governing the relationships and facts within the graph. One such example is the rule $uncle(A, C) = brother(A, B) \text{ and } parent(B, C)$, which describes a family relationship. Prominent examples of KGs include Freebase (Bollacker et al., 2008), YAGO (Suchanek et al., 2007), and DBpedia (Auer et al., 2007). KGs have become integral to numerous applications, ranging from semantic search (Xiong et al., 2017) to recommender systems (Wang et al., 2018). However, a significant challenge in working with KGs is their inherent incompleteness (Destandau and Fekete, 2021) and the presence of noise, as some edges may be incorrect. The task of link prediction addresses this issue and aims to infer new edges between entities in the graph. Link prediction can be approached in either an inductive or transductive setting. In the inductive case, the model performs inference on new, unseen graphs, while in the transductive case, it predicts new links between nodes that were present during training.

Several research fields tackle the task of link prediction. *Symbolic methods* (Eiter et al., 2000; Richardson and Domingos, 2006; Jackson, 1998), grounded in rule-based systems, expert knowledge, and logical reasoning, are employed for this task. Symbolic approaches guarantee logical consistency, but they often suffer from sensitivity to noise. If facts in the KG are actually erronous,

the application of symbolic methods may lead to error propagation. Another major limitation of symbolic methods is scalability. Additionally, *subsymbolic methods* such as graph neural networks (GNNs) (Scarselli et al., 2008) or knowledge graph embeddings (KGE) (Ali et al., 2022) approach link prediction by capturing the structure of the graph in the continuous vector space. These subsymbolic methods show robustness to noise and scalability, but the logical consistency of the model predictions with the rules in the ontology is often not guaranteed.

In recent years, *neurosymbolic methods* have gained attention for their potential to combine the strengths of symbolic reasoning and embedding learning, seeking a balance between logical expressiveness and scalability. Neurosymbolic methods aim to bridge the gap between reasoning accuracy and efficient computation, making them particularly attractive for complex link prediction tasks on large KGs. The field of neurosymbolic AI for link prediction is growing rapidly, with contributions from different research areas. As a result, methods differ in how they integrate symbolic and subsymbolic components, and the logic they can express. Neurosymbolic methods for link prediction are often evaluated independently, each on separate benchmarks and with different baselines, making direct comparisons of methods across the field difficult (Vermeulen et al., 2023; Ott et al., 2023).

Previous work that resemble can be divided into two categories: papers that focus on benchmarking link prediction methods, but without considering neurosymbolic approaches (Liu et al., 2023; Li et al., 2023a), and surveys that discuss neurosymbolic methods for link prediction, but without conducting empirical evaluations (Cheng et al., 2024; DeLong et al., 2025; Feldstein et al., 2024; Wang et al., 2025; Sarker et al., 2021; Vermeulen et al., 2023; Zhang et al., 2024)

**Contribution.** This paper surveys neurosymbolic approaches to link prediction on knowledge graphs and makes the following contributions. First, we provide a qualitative taxonomy of state-of-the-art methods in the field. Second, and in contrast to recent survey papers (Cheng et al., 2024; DeLong et al., 2025; Feldstein et al., 2024; Wang et al., 2025; Sarker et al., 2021; Vermeulen et al., 2023; Zhang et al., 2024), we leverage this taxonomy to conduct a rigorous experimental benchmarking study. All experiments are performed under identical conditions, using the same hardware and preprocessing steps, in order to ensure meaningful comparisons in terms of precision and execution time. This study enables a fair and controlled comparison of neurosymbolic approaches across tasks of increasing complexity. Our evaluation systematically varies several critical factors, including the choice of datasets, the number and composition of logical rules, and the proportion of incorrect rules within the rule sets. This work contributes to the growing field of neurosymbolic link prediction by providing experimental comparisons that may assist researchers in selecting suitable approaches. Our results highlight trade-offs between accuracy, efficiency, and robustness, and reveal key open challenges for future research.

## 2. State-of-the-art

We categorize state-of-the-art link prediction methods for knowledge graphs (KGs) into *symbolic*, *subsymbolic* and *neurosymbolic* methods. Neurosymbolic methods can be further divided into three distinct categories. We classify them as follows: (1) *symbolic methods with learning*, (2) *subsymbolic methods with logical constraints*, and (3) *bidirectional neurosymbolic integration*. The categories and the classified approaches in this work are summarized in Figure 1. In the following sections, we introduce some relevant methods for each category.

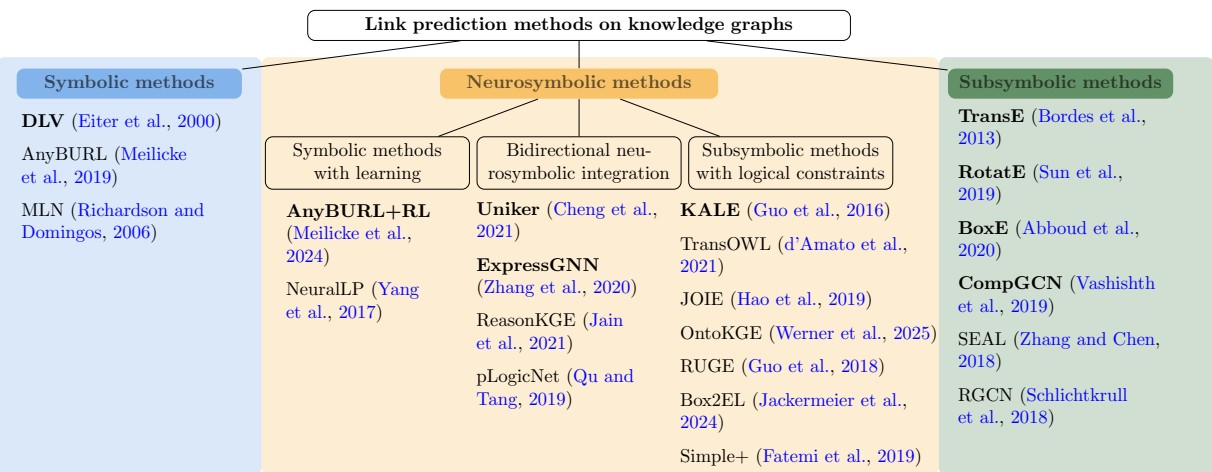

Figure 1: Overview of the methods reviewed in this work. Methods experimentally compared in this work are shown in bold.

## 2.1. Symbolic Methods

Symbolic methods rely on logical reasoning to infer new links in a KG. Methods in this category operate by applying logical rules to derive new facts, ensuring consistency with the given ontology. These rules are typically expressed in first-order logic or its restricted forms, such as Horn rules. A Horn rule is defined as $\eta \leftarrow B(\beta_1 \wedge \ldots \wedge \beta_n)$, where the head $\eta$ is a singular atomic formula and the body $B$ is a conjunction of atomic formulae $\beta_1 \wedge \ldots \wedge \beta_n$. A common approach in symbolic reasoning is *forward chaining*, where inference starts from known facts and iteratively applies rules to infer new facts until no more new facts can be derived. An example of such a method is *DLV* (Eiter et al., 2000) that is based on Datalog (Abiteboul et al., 1995). Some symbolic methods can incorporate *soft rules*, where each rule is associated with a weight representing the confidence in the rule. This allows for a degree of uncertainty in rule application, making the system more robust to noisy or incomplete data. An example of a method using soft rules is *Markov Logic Networks (MLN)* (Richardson and Domingos, 2006), which combine probabilistic graphical models with first-order logic to capture both uncertainty and relational structure. Other methods do not require a predefined set of rules. Instead, they can learn rules directly from the training data. One such method is *AnyBURL* (Meilicke et al., 2019), which learns Horn rules by randomly sampling paths in the knowledge graph, generalizing them into logical rules, and scoring them based on their predictive performance.

## 2.2. Subsymbolic Methods

There are two main categories of subsymbolic methods for link prediction on graphs: Knowledge graph embeddings (KGE) and graph neural networks (GNN). The goal of KGE methods is to learn a dense representation of the KG in a continuous low-dimensional vector space that captures the structural properties of the graph. In KGEs, entities and relations are encoded as lookup matrices. In other words, given a KG $\mathbf{K} = (\mathcal{E}, \mathcal{R}, \mathcal{F})$ with entities $\mathcal{E}$, relations $\mathcal{R}$ and facts $\mathcal{F}$, each entity and relation is represented as a $d$-dimensional embedding vector. These embeddings are learned through an optimization task. In this way, a fact $(head, relation, tail)$ is represented in the vector space as $(\mathbf{h}, \mathbf{r}, \mathbf{t})$, where $\mathbf{h}$ and $\mathbf{t}$ are the vector representations of the head and tail entities. The entity and relation embeddings are randomly initialised at the beginning of the training and optimised

with gradient descent. As a result, embeddings capture the structural representation of entities and relations in the training graph. A *score function* $f : \mathcal{E} \times \mathcal{R} \times \mathcal{E} \to \mathbb{R}$ is defined that takes as input the entity and relation embeddings of a fact $(\mathbf{h}, \mathbf{r}, \mathbf{t})$ and returns its *plausibility*. The exact definition of the score function depends on the design of the KGE model. Given two facts $(h_1, r_1, t_1)$ and $(h_2, r_2, t_2)$, the first is considered more plausible than the second if $f(h_1, r_1, t_1) > f(h_2, r_2, t_2)$. To predict a missing link for a given entity and relation, the model assigns a score to each possible entity in the graph and selects the one with the highest score as the most plausible tail (or head) entity. Prominent knowledge graph embedding models are for example *TransE* (Bordes et al., 2013), *RotatE* (Sun et al., 2019), *BoxE* (Abboud et al., 2020) and *DistMult* (Yang et al., 2015). A broader overview of current KGE methods can be found in (Ali et al., 2022).

GNNs (Scarselli et al., 2008) iteratively refine node representations by leveraging the graph structure through message-passing layers. At each iteration, a node updates its embedding by aggregating information from its neighbors, effectively capturing the local topology and feature context. To derive link prediction scores from the obtained node representations, GNNs for link prediction are combined with a composition operator, usually taken from KGE methods such as DistMult. GNN methods for link prediction include, for example, *RGCN* (Schlichtkrull et al., 2018), *SEAL* (Zhang and Chen, 2018) and *CompGCN* (Vashishth et al., 2019). A broader overview of GNN methods can be found in (Wu et al., 2021).

### 2.3. Neurosymbolic methods

#### 2.3.1. Symbolic systems with learning

Methods in this category are primarily symbolic but consist of a subsymbolic component with a learning stage that plays a supportive role. Auxiliary components such as rule weights, scoring heuristics, ranking functions, or selection strategies are learned or improved using subsymbolic models. The symbolic component in itself is not modified. Inference involves reasoning on the basis of logical rules with or without rules weights, but its execution or guidance is influenced by subsymbolic component. One example of a method in this category is an *AnyBURL + RL* (Meilicke et al., 2024), an extension of AnyBURL which uses reinforcement learning (RL) to improve the efficiency of guided path sampling. While the core component of *AnyBURL + RL* is symbolic, it is complemented by a subsymbolic learning component, which places it within this category. Another method in this category is *Neural LP* (Yang et al., 2017), a neurosymbolic model that learns first-order logical rules. A neural control system learns the rules and their confidence values. Then, during inference, Neural LP uses the logical rules and their weights to derive predictions.

#### 2.3.2. Subsymbolic methods with logical constraints

The methods in this category are essentially subsymbolic, but they are trained using symbolic constraints, such as logical rules, ontologies or type systems, to regularize and guide the learning process. The core representation is subsymbolic, but symbolic knowledge shapes or constrains it. In *KALE* (Guo et al., 2016), the facts of the KG are interpreted as grounded atomic formulae, where all variables are replaced by constants. Based on this, KALE introduces constraints to KGEs by jointly embedding the grounded formulae in first-order logic with the facts of a KG in the same latent space. KALE uses TransE to compute scores for the grounded atomic formulae. To represent complex formulae, KALE encodes logical operators in fuzzy logic. The higher a truth value, the better a formula is satisfied. In the following, a global loss can be defined over the satisfaction of the grounded formulae in fuzzy logic and the atomic formulae as facts in the graph. This way, KALE's loss function penalizes solutions that do not satisfy the logical formulae.

Other methods that fall in this category are for example *TransOWL* (d'Amato et al., 2021) that adds symmetry, equivalence or inversion rules to TransE in form of a geometric regularization term. *JOIE* (Hao et al., 2019) incorporates domain and range constraints of relations by encouraging entity embeddings to be close to the corresponding domain and range embeddings in the vector space. *RUGE* (Guo et al., 2018) and *OntoKGE* (Werner et al., 2025) mine additional facts with rules during the training process and optimize a global loss over explicit and implicit facts. Some KGE models such as *BoxE with rule injection* (Abboud et al., 2020), *Box2EL*(Jackermeier et al., 2024) or *Simple+* (Fatemi et al., 2019) allow for the explicit injection of rules commonly found in ontologies in their geometric model.

### 2.3.3. Bidirectional neurosymbolic integration

Methods in this category feature a tight integration between symbolic and neural components. These components are co-dependent, jointly trained, and often bidirectionally connected. Learning and reasoning continually influence one another, with subsymbolic and symbolic modules either jointly optimized or engaged in a tightly coupled feedback loop.

A method in this category is *UniKer* (Cheng et al., 2021) that integrates a KGE method with a reasoner in an iterative manner to infer implicit facts. KGEs are used to score existing facts and filter out the least plausible proportion of facts. The reasoner then applies Horn rules to infer new facts. This allows UniKer to be robust to noise and to avoid the propagation of contradictions by the reasoner. Another method in this category is *pLogicNet* (Qu and Tang, 2019), which integrates MLNs and KGEs and is optimized using a variational EM algorithm, where the variational distribution is parameterized by a KGE model. During the E-step, the model infers missing facts using the KGE, while in the M-step, it updates the weights of the logical rules based on both observed and predicted triples. This integration allows the model to reason under uncertainty while effectively combining symbolic rules with the information captured by embeddings and observed data. As an extension of pLogicNet, *ExpressGNN* (Zhang et al., 2020), uses GNNs instead of flattened embedding tables as entity representation in order to explicitly capture graph structure. Since the EM framework iteratively leverages the symbolic and the subsymbolic component, ExpressGNN and pLogicNet are placed in this category.

Other methods in this category include, for example, *RulE* (Tang et al., 2024) and *ReasonKGE* (Jain et al., 2021). Neural probabilistic programming approaches (Manhaeve et al., 2018; Li et al., 2023b) also fall into this category. However, to the best of our knowledge, they have not yet been applied to the link prediction setting considered here. We have also not succeeded in leveraging their architectures based on weighted model counting in a way that scales to the size of the knowledge graphs involved. This leaves open the opportunity to investigate whether they can be effectively applied in this context.

## 3. Experimental Methodology

The goal of this experimental methodology is to evaluate neurosymbolic methods from the categories in Section 2, while also including representative symbolic and subsymbolic approaches as reference points, with respect to the following questions:

**Q1** How *accurate* are the link prediction results of these methods?

**Q2** Are the methods *scalable*?
  **Q2a** Do they scale with the *size of the graph*?
  **Q2b** Do they scale with the *number of rules*?

**Q3** How does the *level of provided knowledge* impact the link prediction accuracy of methods?

| Datasets | Entities | Relations | Triples | Train | Test | Valid | Rules |
|---|---|---|---|---|---|---|---|
| Family_small | 3,007 | 12 | 10,741 | 5,868 | 2,835 | 2,038 | 41 |
| Family_medium | 3,007 | 12 | 28,356 | 19,845 | 5,681 | 2,830 | 41 |
| WN18RR | 40,943 | 11 | 93,003 | 86,835 | 3,134 | 3,034 | 8 |
| FB15K237 | 14,541 | 237 | 310,116 | 272,115 | 20,466 | 17,535 | 507 |

Table 1: Overview of the datasets.

**Q4** Are the methods *robust* to incorrect knowledge?

### 3.1. Tasks

To provide insights into these questions, an experimental evaluation of the methods is conducted on the following link prediction tasks. Statistics on the datasets are provided in Table 1.

**Family.** In the Family dataset (Denham, 1973), entities in the graph represent people, while relations correspond to their kinship ties. The task is to predict missing relationships between entities. We use a set of rules 41 rules that have been automatically mined using AMIE+ (Galárraga et al., 2015; Cheng et al., 2021). The set of rules contains Horn rules, such as daughter(x,y) ← sister(x,z) ∧ son(z,y). We consider two versions of the Family dataset that share the same unique entities and relations but differ in size: Family_small (Han et al., 2023) and Family_medium (Cheng et al., 2021).

**WN18RR.** In WN18RR, words are modeled as entities, with relationships expressing the lexical links between them. WN18RR is a dataset extracted from WordNet and is a widely used benchmark knowledge graph datasets introduced in (Bordes et al., 2013). Unlike the Family dataset, where relationships are defined by strict logical rules, the rules for Wordnet are soft rules, meaning they only hold true with a certain confidence. We took the rule set used in (Cheng et al., 2021) and keep only the rule with a confidence weight above 0.3. As example, 0.4: synset_domain_topic_of(x,z)← hypernym(x,y) ∧ synset_domain_topic_of(y,z) means that if x is a more specific concept (a hyponym) of y, and y is associated with a domain z, then x is likely to be also associated with the same domain z.

**FB15k237.** The FB15k237 dataset (Toutanova and Chen, 2015), extracted from Freebase (Bollacker et al., 2008), is another widely used large-scale benchmarks for link prediction, restricted to 237 relations. Entities represent real-world concepts such as people, places, and movies, while relations define how these entities are connected. As prior knowledge, we use the soft rules provided in (Zhang et al., 2020), extracted with NeuralLP (Yang et al., 2017), containing 509 rules. For instance, the rule 0.7: /film/film/genre ←/film/film/prequel_v ∧ /film/film/genre suggests that a movie shares the same genre as its prequel, with a confidence weight of 0.7.

### 3.2. Experimental setting

From each category of methods introduced in Section 2, at least one representative approach is selected. The tested approaches are symbolic (DLV) (Eiter et al., 2000), sub-symbolic (TransE (Bordes et al., 2013), RotatE (Sun et al., 2019), BoxE (Abboud et al., 2020), CompGCN (Vashishth et al., 2019)) and neurosymbolic (KALE (Guo et al., 2016), ExpressGNN (Zhang et al., 2020), Uniker (Cheng et al., 2021), AnyBURL (Meilicke et al., 2024)), as represented in Figure 1. As several methods only work in a transductive setting, all methods are tested in a transductive setting to ensure a fair comparison. In each test, each method is executed five times, with the same conditions and hyperparameters, but different random seeds. Implementation details such

as hyperparameters, data preprocessing steps and hardware details are given in Appendix B. The code is publicly available[1].

**Q1: Link prediction accuracy.**    To compare the accuracy of the link prediction methods, we use the common rank-based metrics MRR, HITS@1, HITS@3, and HITS@10. Each method is trained on the training set and the metrics are computed on the test set for each dataset. Train, valid and test parts of each dataset are separated as they are in the first version of the dataset: (Han et al., 2023) for family_small, (Cheng et al., 2021) for family_medium, (Bordes et al., 2013) for WN18RR and (Toutanova and Chen, 2015) for FB15k237.

**Q2: Scalability.**    The scalability analysis is based on the total measured execution times for training and testing. To ensure comparability, experiments are carried out using the same machine and identical conditions. A timeout of 48 hours is set for FB15k237 and WN18RR, while a shorter timeout of 15 hours is used for Family_small and Family_medium, given their smaller size compared to the other datasets. For **Q2a**, we compare the execution time of all methods across each dataset. The experiments on FB15k237 and WN18RR are particularly informative for assessing scalability, due to the larger size of these datasets. We analyze both the training and testing time, as some applications may prioritize fast training, while others require quick inference during testing. To analyse scalability with respect to the number of provided rules (**Q2b**), we consider five rule sets of different size (41, 33, 25, 16, 8) for the dataset Family_small[2]. In the case of AnyBURL+RL, the training time can be configured by the user. To ensure a fair comparison, we tested multiple training durations and selected the shortest time that did not affect the quality of the predictions. This resulted in a training time of 100 seconds.

**Q3: Link prediction accuracy for different levels of provided knowledge.**    To evaluate how the availability of provided knowledge affects the different methods, we evaluate the methods on Family_small with rule sets of different sizes (8, 16, 25, 33 and 41), as defined in Q2a.

**Q4: Robustness.**    To study the robustness of the methods to ontologies that potentially contain rules that are not consistent with the data, we corrupt the rules. On Family_small, we compare five different rule sets with an increasing proportion of corrupted rules (0%, 20%, 40%, 60% and 80%). For instance, for 40% of corrupted knowledge, the rule set contains 25 correct rules and 16 corrupted rules. These corrupted rules are randomly generated and syntactically valid but semantically incorrect, for example `brother(x,y) ← aunt(x,z) ∧ mother(z,y)`.

## 4. Results

**Q1: Link prediction accuracy.**    Figure 2 visualizes the MRR, which indicates how well they predict new links. Table 2 in Appendix A shows the average rank-based metrics (MRR, HITS@1, HITS@3, HITS@10) and standard deviations of all methods on all datasets in this work. On most datasets, the neurosymbolic methods achieve the highest accuracy, followed by subsymbolic methods, while the symbolic method tends to be the least accurate. The three best performing methods are ExpressGNN on FB15k237 and Uniker on Family_small and AnyBURL+RL on WN18RR and Family_medium. Moreover, while ExpressGNN achieves the best overall performance on FB15k237, its performance is comparatively lower on the smaller datasets. This may be due to the fact that parameter-intensive GNNs like ExpressGNN require a sufficient amount of training data, which is more readily available in larger datasets such as FB15k237. For WN18RR, the methods that rely

---

1. https://gitlab.inria.fr/tyrex-public/nesy_link_prediction_benchmark
2. Notice that this does not make any difference for subsymbolic and AnyBURL methods which do not use rules.

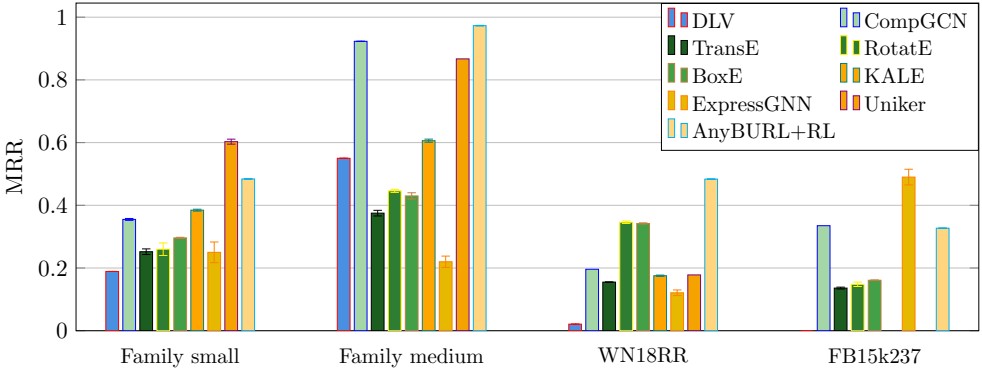

Figure 2: Link prediction results (average MRR) for the compared methods. DLV, KALE and Uniker timeout on FB15k237. Bar order follows legend. Green tones stand for subsymbolic, orange tones for neurosymbolic and blue for symbolic methods. Error bars indicate standard deviations.

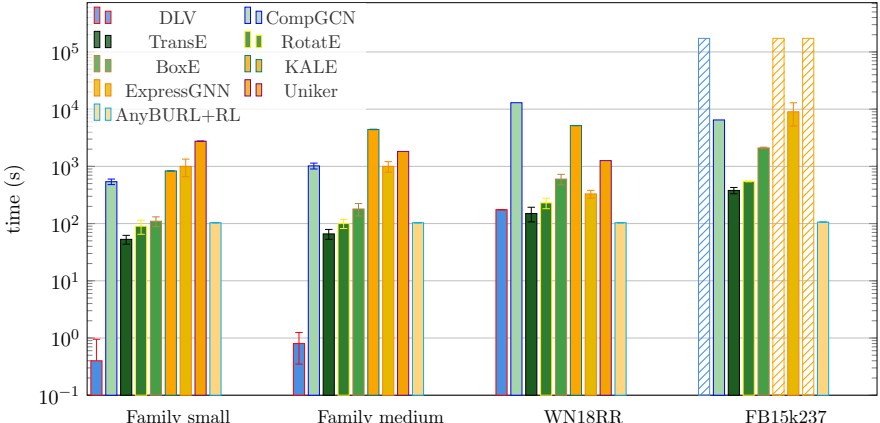

Figure 3: Average training time (log scale, seconds). Hatched bars indicate timeouts (DLV, KALE, Uniker on FB15k237). Bar order follows legend. Green tones stand for subsymbolic, orange tones for neurosymbolic and blue for symbolic methods.

directly on a provided rule set — DLV, Uniker, KALE, and ExpressGNN — exhibit particularly low MRR scores. This is likely due to the small size of the rule set, which contains only eight rules. This limitation is especially impactful for methods that depend on external rule knowledge. Regarding the standard deviation, ExpressGNN is slightly unstable, exhibiting a higher standard deviation than the other methods.

**Q2a: Scalability with respect to the graph size.** Figure 3 visualizes the total training time of all methods across datasets. Table 3 (in Appendix A) contains the full comparison of the execution time for training and testing for all methods and datasets. AnyBURL+RL, with the training time limits chosen in this work, scales best. Among the remaining methods, subsymbolic methods scale best by far and consistently outperform neurosymbolic approaches in speed. The symbolic algorithm, DLV, is the most efficient on small datasets but results in a timeout on the larger FB15k237 dataset. Among neurosymbolic methods Uniker and KALE timeout on FB15k237. Only ExpressGNN scales to the size of FB15k237. However, during inference, ExpressGNN is considerably slower than all other methods, with TransE, RotatE, and CompGCN being the fastest.

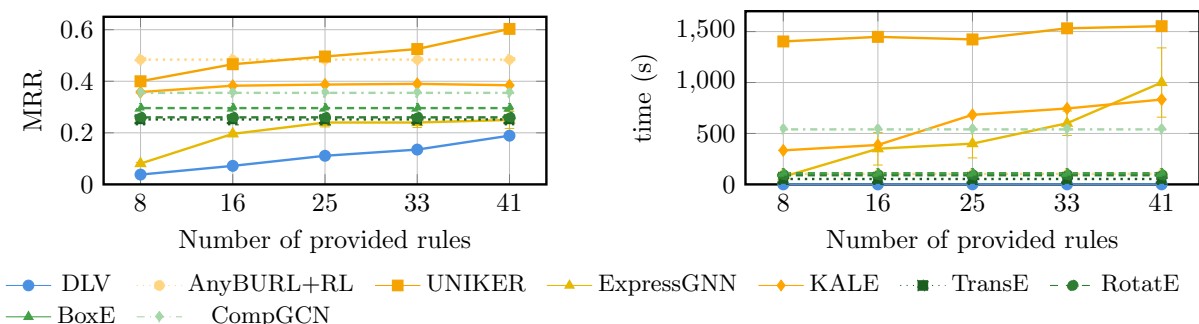

Figure 4: Left: Average MRR vs. number of provided rules on Family_small. Right: Execution time in seconds vs. number of provided rules on Family_small.

**Q2b: Scalability with respect to the number of rules.** Figure 4 (right) shows the execution times for an increasing number of provided rules on Family_small. For both ExpressGNN and KALE, training time increases as the number of rules in the knowledge base grows, with training time increasing by a factor of 7 for ExpressGNN and by a factor of 2 for KALE. In contrast, the training time of Uniker decreases as rules are added. This behavior may be due to the fact that Uniker generates new facts during training. When there are fewer rules, it takes longer to produce the necessary facts for learning. In contrast, when the graph is more complete, fewer new facts need to be generated, enabling the model to converge more quickly. This demonstrates that all methods training time is notably influenced by the given ontology.

**Q3: Link prediction accuracy for different levels of provided knowledge.** Figure 4 (left) shows the MRR for different numbers of rules. As expected, the MRR improves substantially as the number of rules increases for the symbolic method DLV. Subsymbolic methods remain unaffected since they disregard prior knowledge and show a constant MRR. For neurosymbolic algorithms, the impact varies. For KALE, which is placed in category of subsymbolic with symbolic constraints that are closer to symbolic methods, only a small improvement in MRR is observed as the number of rules increases. In contrast, Uniker and ExpressGNN show a more pronounced positive trend, indicating that the availability of rules contributes to higher MRR for these neurosymbolic methods. This finding supports the observation made in Q1, where the neurosymbolic methods ExpressGNN, Uniker, and KALE show reduced performance on WN18RR likely due to the limited number of available rules for this dataset.

**Q4: Robustness.** The results for robustness to incorrect rules are shown in Figure 5. As expected, the symbolic method DLV faces computational instability when incorrect rules are intro-

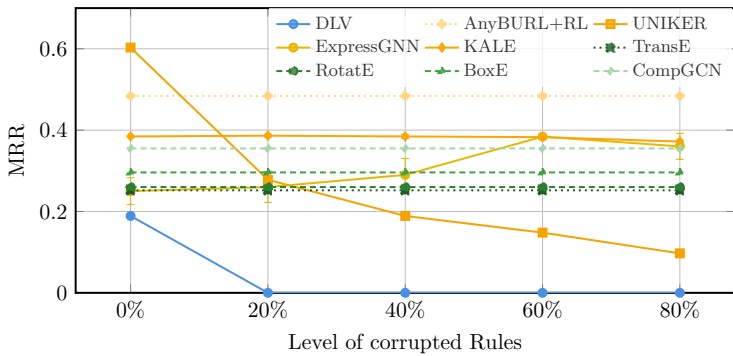

Figure 5: MRR vs. proportion of corrupted rules on the Family_small dataset.

duced and does thus not tolerate any inconsistency. This is demonstrated by a timeout occurring despite the small dataset size when the rule set contains only 20% corrupted rules. Subsymbolic methods, on the other hand, remain unaffected since they do not rely on the ontology. For neurosymbolic methods, the impact varies. KALE's performance remains stable despite increasing proportions of corrupted rules, suggesting that its foundation on a subsymbolic method contributes to its robustness. In contrast, the accuracy of Uniker declines substantially as the rule quality deteriorates. Interestingly, ExpressGNN's MRR increases when the quality of the logic rules is reduced. This counterintuitive behavior may be explained by the fact that ExpressGNN, with its GNN containing many trainable parameters, is particularly susceptible to overfitting on the relatively small Family_small dataset. In this context, some incorrect rules could act as a form of regularization, thereby improving performance.

## 5. Conclusion and Perspectives

This work introduced a taxonomy of neurosymbolic approaches for link prediction and conducted a systematic benchmarking of representative methods. Our results reveal a much more nuanced picture than commonly assumed.

First, when considering accuracy within the allowed time limits, a clear trend emerges depending on the type of dataset. On small, synthetic datasets (e.g., Family small and medium), neurosymbolic methods consistently outperform subsymbolic baselines. However, this advantage collapses as we move toward larger, more realistic benchmarks such as WN18RR and FB15k237. On these, the performance of neurosymbolic approaches often deteriorates significantly, both in terms of prediction quality and execution time, calling into question their scalability and robustness. In some cases, neurosymbolic methods perform worse than their subsymbolic counterparts, even on accuracy metrics, and several approaches fail to complete within generous time limits (notably on FB15k237). Surprisingly, the method that performed the worst on the small synthetic datasets emerges as the most robust on the larger, more challenging one.

In terms of execution time, the conclusion is unambiguous: neurosymbolic methods are consistently more computationally expensive. The only exception is AnyBURL+RL, which always completes within a fixed 100-second timeout, owing to its interruptible design. This points to a promising direction for future work: designing neurosymbolic systems to be interruptible and incremental, capable of producing usable results within a configurable time budget. While this may not apply to all approaches, it could be essential for ensuring practical scalability and avoiding exclusion from real-world applications. AnyBURL+RL further illustrates that even within limited time constraints, it is possible to outperform the accuracy of subsymbolic baselines, highlighting the potential of well-engineered incremental neurosymbolic designs.

Finally, beyond scalability, our study reveals that some neurosymbolic models are more sensitive then others to ontology quality. While leveraging logical rules can enhance accuracy, it also introduces vulnerabilities when rules are incomplete or incorrect, which is common in practice. Unlike subsymbolic methods, which are immune to such issues, neurosymbolic approaches may degrade. Their effectiveness thus hinges not only on symbolic integration but also on the ability to assess and handle imperfect knowledge.

Overall, our findings suggest that conclusions drawn from isolated evaluations may not fully capture the practical strengths and limitations of neurosymbolic approaches. This underscores the need for more systematic and fair benchmarking efforts, such as the one undertaken in this study.

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

# Appendix A. Experimental Results

This section provides additional presentations of the results. Table 2 shows the average rank-based metrics and standard deviations of all methods on all datasets. The training and testing time of all methods on all datasets are presented in Table 3. Tabls 3 summarizes all training and testing

| | Method | Family_small | Family_medium | WN18RR | FB15k237 |
|---|---|---|---|---|---|
| **Symbolic** | DLV | MRR: 0.189 (0.0)
HITS@1: 0.042 (0.0)
HITS@3: 0.253 (0.0)
HITS@10: 0.397 (0.0) | MRR: 0.550 (0.0)
HITS@1: 0.178 (0.0)
HITS@3: 0.751 (0.0)
HITS@10: 0.914 (0.0) | MRR: 0.021 (0.0)
HITS@1: 0.012 (0.0)
HITS@3: 0.024 (0.0)
HITS@10: 0.027 (0.0) | TIMEOUT |
| **Subsymbolic** | TransE | MRR: 0.252 (0.009)
HITS@1: 0.094 (0.008)
HITS@3: 0.302 (0.011)
HITS@10: 0.634 (0.016) | MRR: 0.375 (0.009)
HITS@1: 0.171 (0.010)
HITS@3: 0.498 (0.009)
HITS@10: 0.77 (0.015) | MRR: 0.155 (0.001)
HITS@1: 0.019 (0.001)
HITS@3: 0.235 (0.004)
HITS@10: 0.413 (0.004) | MRR: 0.136 (0.003)
HITS@1: 0.085 (0.003)
HITS@3: 0.144 (0.003)
HITS@10: 0.237 (0.004) |
| | RotatE | MRR: 0.26 (0.02)
HITS@1: 0.12 (0.014)
HITS@3: 0.31 (0.025)
HITS@10: 0.59 (0.031) | MRR: 0.446 (0.005)
HITS@1: 0.263 (0.008)
HITS@3: 0.562 (0.004)
HITS@10: 0.800 (0.002) | MRR: 0.346 (0.004)
HITS@1: 0.281 (0.004)
HITS@3: 0.380 (0.006)
HITS@10: 0.469 (0.004) | MRR: 0.148 (0.007)
HITS@1: 0.090 (0.006)
HITS@3: 0.162 (0.007)
HITS@10: 0.26 (0.01) |
| | BoxE | MRR: 0.296 (0.002)
HITS@1: 0.150 (0.002)
HITS@3: 0.348 (0.005)
HITS@10: 0.650 (0.007) | MRR: 0.43 (0.010)
HITS@1: 0.25 (0.014)
HITS@3: 0.54 (0.011)
HITS@10: 0.781 (0.008) | MRR: 0.342 (0.002)
HITS@1: 0.275 (0.004)
HITS@3: 0.374 (0.002)
HITS@10: 0.472 (0.001) | MRR: 0.161 (0.001)
HITS@1: 0.102 (0.002)
HITS@3: 0.173 (0.002)
HITS@10: 0.274 (0.002) |
| | CompGCN | MRR: 0.355 (0.003)
HITS@1: 0.210 (0.003)
HITS@3: 0.426 (0.004)
HITS@10: 0.678 (0.003) | MRR: 0.923 (0.0)
HITS@1: 0.877 (0.01)
HITS@3: 0.969 (0.0)
HITS@10: 0.980 (0.0) | MRR: 0.196
Hit@1: 0.054
Hit@3: 0.281
Hit@10: 0.518 | MRR: 0.335
HITS@1: 0.245
HITS@3: 0.369
HITS@10: 0.513 |
| **Neurosymbolic** | KALE | MRR: 0.424 (0.003)
Hit@1: 0.232 (0.002)
Hit@3: 0.534 (0.003)
Hit@10: **0.836** (0.002) | MRR: 0.612 (0.005)
Hit@1: 0.453 (0.009)
Hit@3: 0.731 (0.005)
Hit@10: 0.867 (0.002) | MRR: 0.188 (0.002)
Hit@1: 0.037 (0.002)
Hit@3: 0.277 (0.005)
Hit@10: 0.468 (0.002) | TIMEOUT |
| | ExpressGNN | MRR: 0.25 (0.033)
Hit@1: 0.15 (0.033)
Hit@3: 0.26 (0.040)
Hit@10: 0.53 (0.027) | MRR: 0.22 (0.018)
Hit@1: 0.15 (0.015)
Hit@3: 0.24 (0.021)
Hit@10: 0.35 (0.024) | MRR: 0.121 (0.009)
Hit@1: 0.068 (0.004)
Hit@3: 0.13 (0.012)
Hit@10: 0.24 (0.023) | MRR: **0.49** (0.025)
Hit@1: **0.43** (0.029)
Hit@3: **0.53** (0.025)
Hit@10: **0.61** (0.017) |
| | AnyBURL+RL | MRR: 0.484 (0.001)
Hit@1: 0.323 (0.003)
Hit@3: 0.592 (0.001)
Hit@10: 0.812 (0.001) | MRR: **0.973** (0.001)
Hit@1: **0.965** (0.002)
Hit@3: **0.980** (0.000)
Hit@10: **0.984** (0.000) | MRR: **0.484** (0.001)
Hit@1: **0.448** (0.001)
Hit@3: **0.498** (0.001)
Hit@10: **0.557** (0.001) | MRR: 0.327 (0.001)
Hit@1: 0.243 (0.001)
Hit@3: 0.357 (0.001)
Hit@10: 0.499 (0.001) |
| | Uniker | MRR: **0.603** (0.008)
Hit@1: **0.50** (0.010)
Hit@3: **0.673** (0.009)
Hit@10: 0.814 (0.003) | MRR: 0.867
Hit@1: 0.797
Hit@3: 0.927
Hit@10: 0.978 | MRR: 0.178
Hit@1: 0.006
Hit@3: 0.308
Hit@10: 0.465 | TIMEOUT |

Table 2: Link prediction results. The metrics are average values over five runs. The highest values per dataset are marked in bold. Standard deviations are given in brackets.

times. Figure 6 visualizes the testing times of all methods as a barchart.

# Appendix B. Implementation Details

The code for the experiments is obtained from the GitHub repositories provided in the respective papers KALE[3], ExpressGNN [4], Uniker[5] and CompGCN[6]. For the subsymbolic KGE methods, we use the Pykeen library (Ali et al., 2021). For AnyBURL+RL, the code is taken from their

---

3. https://github.com/iieir-km/KALE

4. https://github.com/expressGNN/ExpressGNN

5. https://github.com/vivian1993/UniKER

6. https://github.com/malllabiisc/CompGCN

|  | Method | Family_small | Family_medium | WN18RR | FB15k237 |
|---|---|---|---|---|---|
| Symbolic | DLV | Training: 0.4s (0.55)
Inference: 8.7s (0.17) | Training: 0.8s (0.45)
Inference: 49s (1.2) | Training: 175s (0.8)
Inference: 5h59 | TIMEOUT |
| Sub-
sym-
bolic | TransE | Training: 53s (9.3)
Inference: 0.1s (0.0) | Training: 66s (13)
Inference: 0.2s (0.0) | Training: 150s (43)
Inference: 0.9s (0.0) | Training: 380s (48)
Inference: 3.1s (0.4) |
|  | RotatE | Training: 90s (25)
Inference: 0.2s (0.0) | Training: 100s (18)
Inference: 1.1s (0.0) | Training: 230s (48)
Inference: 3.0s (0.0) | Training: 545s (8.5)
Inference: 5.9s (0.0) |
|  | BoxE | Training: 110s (21)
Inference: 1.8s (0.0) | Training: 180s (44)
Inference: 4.3s (0.0) | Training: 10 min (125)
Inference: 20.9s (0.2) | Training: 35 min (65)
Inference: 47.3s (0.2) |
|  | CompGCN | Training: 540s (60)
Inference: 0.9s (0.0) | Training: 17 min 00s (120)
Inference: 1.28s (0.0) | Training: 3h 36min
Inference: 3s | Training: 1h48 min
Inference: 8.9s |
| Neuro-
sym-
bolic | KALE | Training: 13 min 53s (5.4)
Inference: 4s (0.0) | Training: 1h 14min (40)
Inference: 9.2s (0.4) | Training: 1h 26min (20)
Inference: 62s (1.1) | TIMEOUT |
|  | ExpressGNN | Training: 16 min 40s (340)
Inference: 41.4s (0.6) | Training: 16 min 40s (209)
Inference: 84.6s (0.6) | Training: 330s (51)
Inference: 300s (14) | Training: 2h 30min (3930)
Inference: 300s (4.1) |
|  | AnyBURL+RL | Training: 103s (0.0)
Inference: 41.6s (1.5) | Training: 103.4s (0.5)
Inference: 140s (5.5) | Training: 103.4s (0.5)
Inference: 9s (0.0) | Training: 106.2s (4.5)
Inference: 76s (0.0) |
|  | Uniker | Training: 25 min 53s (12)
Inference: 19s (2.7) | Training: 30 min
Inference: 18.1s (17) | Training: 20 min 01s
Inference: 195s | TIMEOUT |

Table 3: Average execution time for training and testing. The standard deviation over five runs is shown in brackets in seconds.

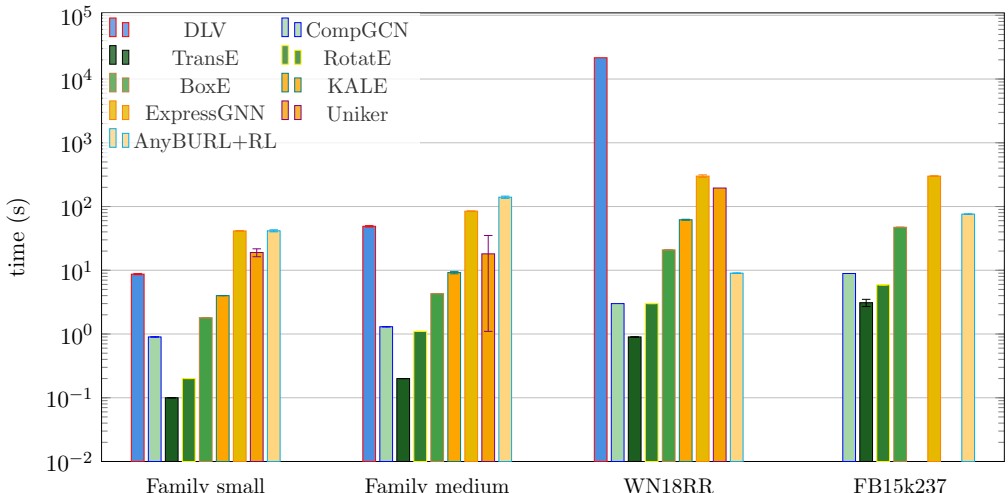

Figure 6: Average testing time (log scale, seconds). DLV, KALE and Uniker timeout on FB15k237. Bar order follows legend. Green tones stand for subsymbolic, orange tones for neurosymbolic and blue for symbolic methods.

website[7]. The code of DLV is taken from the dowloading page[8] of DLV website. The format of the datasets is adjusted to be compatible with all methods. As mentioned in (Liu et al., 2023) and (Li et al., 2023a), hyperparameter choices can significantly influence performance, so we carefully tuned the hyperparameters for each method to ensure a fair comparison. To determine them for all methods, we started with the default values from their respective implementations and made slight

---

7. https://web.informatik.uni-mannheim.de/AnyBURL/

8. https://www.dlvsystem.it/dlvsite/dlv-download/

adjustments to optimize performance within a shorter training time. Further experimental details and a complete list of hyperparameters are provided in Appendix C. We conduct all experiments on a Linux server equipped with an Intel Xeon Silver 4210R CPU (10 cores, 20 threads, 2.40 GHz), 180 GB of system memory, an NVIDIA RTX A6000 GPU with 49 GB of VRAM.

## Appendix C. Hyperparameters and Instructions

We provide the hyperparameters used for each approach and the detailed instructions in form of commands to reproduce the experiments with each method. Additional information can be found in our Gitlab repository at https://gitlab.inria.fr/tyrex-public/nesy_link_prediction_benchmark which will be made public after the review period.

### C.1. DLV

DLV does not contain any hyperparameters to configure.

### C.2. TransE, RotatE and BoxE

The code used for these systems is taken from PyKEEN (Ali et al., 2021). Default hyperparameters provided in the documentation (see https://pykeen.readthedocs.io/en/stable/reference/models.html) and reported in (Ali et al., 2022) have been used.

### C.3. CompGCN

The following commands can be used to run the results with CompGCN.

```
$python3~run.py~-name~best\_model~-score\_func~conve~-opn~corr~-data~family\_small$~\\
$python3~run.py~-name~best\_model~-score\_func~conve~-opn~corr~-data~family\_medium$~\\
$python3~run.py~-score\_func~transe~-opn~sub~-gamma~9~-hid\_drop~0.1~-init\_dim~200~-data
    ~wn18$~\\
$python3~run.py~-score\_func~transe~-opn~sub~-gamma~9~-hid\_drop~0.1~-init\_dim~200~-data
    ~FB15k-237$ \\
```

The following hyperparameters have been used:

- `name`: the name given for the run (for storing model parameters)

- `score_func`: link prediction score function

- `opn`: composition operation used in CompGCN. Possible values:

  - `sub` for subtraction operation: $\Phi(e_s, e_r) = e_s - e_r$
  - `mult` for multiplication operation: $\Phi(e_s, e_r) = e_s \times e_r$
  - `corr` for circular-correlation: $\Phi(e_s, e_r) = e_s \star e_r$

- `gamma`: Margin (default: 40.0)

- `hid_drop`: Dropout after GCN (default: 0.3)

- `init_dim`: Initial dimension size for entities and relations (default: 100)

## C.4. KALE

In the KALE GitHub repository, the hyperparameters $k$ (embedding dimension) and $d$ (margin value) are set to 50 and 0.2 in the provided example. To evaluate KALE on the Family task, we tested multiple values for $k$ (ranging from 20 to 100) and $d$ (ranging from 0.1 to 0.5). The best results, balancing rank-based metrics and training efficiency, were obtained with $k = 80$ and $d = 0.3$. Consequently, we used these values to run KALE and report its performance. KALE is implemented in Java. The following commands have been used to run the experiments.

```
java -jar KALE.jar -train datasets/family-small/train.txt -valid datasets/family-small/
    valid.txt -test datasets/family-small/test.txt -rule datasets/family-small/groundings
    .txt -m 12 -n 2411 -w 0.1 -k 80 -d 0.3 -ge 0.1 -gr 0.1 -# 1500 -skip 50
java -jar KALE.jar -train datasets/family-medium/train.txt -valid datasets/family-medium/
    valid.txt -test datasets/family-medium/test.txt -rule datasets/family-medium/
    groundings.txt -m 12 -n 2968 -w 0.1 -k 80 -d 0.3 -ge 0.1 -gr 0.1 -# 1500 -skip 50
java -jar KALE.jar -train datasets/wn18/train.txt -valid datasets/wn18/valid.txt -test
    datasets/wn18/test.txt -rule datasets/wn18/groundings.txt -m 18 -n 40559 -w 0.1 -k 50
     -d 0.2 -ge 0.1 -gr 0.1 -# 1000 -skip 50
java -Xmx32g -Xms8g -jar KALE.jar -train datasets/fb15k/train.txt -valid datasets/fb15k/
    valid.txt -test datasets/fb15k/test.txt -rule datasets/fb15k/groundings.txt -m 237 -n
     14505 -w 0.1 -k 120 -d 0.3 -ge 0.1 -gr 0.1 -# 1000 -skip 50
```

Listing 1: KALE Commands for Family_small

Here are the commands to test different levels of provided knowledge:

```
java -jar KALE.jar -train datasets/family-small/train.txt -valid datasets/family-small/
    valid.txt -test datasets/family-small/test.txt -rule datasets/family-small/
    groundings_80%.txt -m 12 -n 2411 -w 0.1 -k 80 -d 0.3 -ge 0.1 -gr 0.1 -# 1500 -skip 50
java -jar KALE.jar -train datasets/family-small/train.txt -valid datasets/family-small/
    valid.txt -test datasets/family-small/test.txt -rule datasets/family-small/
    groundings_60%.txt -m 12 -n 2411 -w 0.1 -k 80 -d 0.3 -ge 0.1 -gr 0.1 -# 1500 -skip 50
java -jar KALE.jar -train datasets/family-small/train.txt -valid datasets/family-small/
    valid.txt -test datasets/family-small/test.txt -rule datasets/family-small/
    groundings_40%.txt -m 12 -n 2411 -w 0.1 -k 80 -d 0.3 -ge 0.1 -gr 0.1 -# 1500 -skip 50
java -jar KALE.jar -train datasets/family-small/train.txt -valid datasets/family-small/
    valid.txt -test datasets/family-small/test.txt -rule datasets/family-small/
    groundings_20%.txt -m 12 -n 2411 -w 0.1 -k 80 -d 0.3 -ge 0.1 -gr 0.1 -# 1500 -skip 50
```

Here are the commands to test the robustness of the method against wrong knowledge:

```
java -jar KALE.jar -train datasets/family-small/train.txt -valid datasets/family-small/
    valid.txt -test datasets/family-small/test.txt -rule datasets/family-small/
    groundings_20%_wrong.txt -m 12 -n 2411 -w 0.1 -k 80 -d 0.3 -ge 0.1 -gr 0.1 -# 1500 -
    skip 50
java -jar KALE.jar -train datasets/family-small/train.txt -valid datasets/family-small/
    valid.txt -test datasets/family-small/test.txt -rule datasets/family-small/
    groundings_40%_wrong.txt -m 12 -n 2411 -w 0.1 -k 80 -d 0.3 -ge 0.1 -gr 0.1 -# 1500 -
    skip 50
java -jar KALE.jar -train datasets/family-small/train.txt -valid datasets/family-small/
    valid.txt -test datasets/family-small/test.txt -rule datasets/family-small/
    groundings_60%_wrong.txt -m 12 -n 2411 -w 0.1 -k 80 -d 0.3 -ge 0.1 -gr 0.1 -# 1500 -
    skip 50
```

```
java -jar KALE.jar -train datasets/family-small/train.txt -valid datasets/family-small/
    valid.txt -test datasets/family-small/test.txt -rule datasets/family-small/
    groundings_80%_wrong.txt -m 12 -n 2411 -w 0.1 -k 80 -d 0.3 -ge 0.1 -gr 0.1 -# 1500 -
    skip 50
```

The commands include the following hyperparameters:

- m: number of relations

- n: number of entities

- k: embedding dimensionality

- d: value of the margin

- ge: initial learning rate of matrix E for AdaGrad

- gr: initial learning rate of matrix R for AdaGrad

- #: number of iterations

- skip: number of skipped iterations

### C.5. ExpressGNN

The following commands can be used to run the experiments with ExpressGNN.

```
python3~-m~main.train~-data\_root~data/family\_small~-rule\_filename~rules.txt~-slice\
    _dim~16~-batchsize~16~-embedding\_size~256~-gcn\_free\_size~255~-load\_method~1~-exp\
    _folder~exp~-exp\_name~family\_small -device cuda \\
python3~-m~main.train~-data\_root~data/family\_medium~-rule\_filename~rules.txt~-slice\
    _dim~16~-batchsize~16~-embedding\_size~256~-gcn\_free\_size~255~-load\_method~1~-exp\
    _folder~exp~-exp\_name~family\_medium -device cuda \\
python3~-m~main.train~-data\_root~data/wn18~-rule\_filename~rules.txt~-slice\_dim~16~-
    batchsize~16~-use\_gcn~1~-batchsize~16~-embedding\_size~128~-gcn\_free\_size~127~-
    patience~20~-lr\_decay\_patience~100~-entropy\_temp~1~-load\_method~1~-exp\_folder~
    exp~-exp\_name~wn18 -device cuda \\
python3~-m~main.train~-data\_root~data/fb15k-237~-rule\_filename~cleaned\_rules\_weight\
    _larger\_than\_0.9.txt~-slice\_dim~16~-batchsize~16~-use\_gcn~1~-num\_hops~1~-
    embedding\_size~128~-gcn\_free\_size~127~-patience~20~-lr\_decay\_patience~100~-
    entropy\_temp~1~-load\_method~1~-exp\_folder~exp~-exp\_name~fb15k-237 -device cuda \\
```

Here are the commands to test different levels of provided knowledge:

```
python3~-m~main.train~-data\_root~data/family\_small~-rule\_filename~rules\_exp\_80\%.txt
    ~-slice\_dim~16~-batchsize~16~-embedding\_size~256~-gcn\_free\_size~255~-load\_method
    ~1~-exp\_folder~exp~-exp\_name~family\_small -device cuda \\
python3~-m~main.train~-data\_root~data/family\_small~-rule\_filename~rules\_exp\_60\%.txt
    ~-slice\_dim~16~-batchsize~16~-embedding\_size~256~-gcn\_free\_size~255~-load\_method
    ~1~-exp\_folder~exp~-exp\_name~family\_small -device cuda \\
python3~-m~main.train~-data\_root~data/family\_small~-rule\_filename~rules\_exp\_40\%.txt
    ~-slice\_dim~16~-batchsize~16~-embedding\_size~256~-gcn\_free\_size~255~-load\_method
    ~1~-exp\_folder~exp~-exp\_name~family\_small -device cuda \\
```

```
python3~-m~main.train~-data\_root~data/family\_small~-rule\_filename~rules\_exp\_20\%.txt
    ~-slice\_dim~16~-batchsize~16~-embedding\_size~256~-gcn\_free\_size~255~-load\_method
    ~1~-exp\_folder~exp~-exp\_name~family\_small -device cuda \\
```

Here are the commands to test the robustness of the method against wrong knowledge:

```
python3~-m~main.train~-data\_root~data/family\_small~-rule\_filename~rules\_exp\_20\%\
    _wrong.txt~-slice\_dim~16~-batchsize~16~-embedding\_size~256~-gcn\_free\_size~255~-
    load\_method~1~-exp\_folder~exp~-exp\_name~family\_small -device cuda \\
python3~-m~main.train~-data\_root~data/family\_small~-rule\_filename~rules\_exp\_40\%\
    _wrong.txt~-slice\_dim~16~-batchsize~16~-embedding\_size~256~-gcn\_free\_size~255~-
    load\_method~1~-exp\_folder~exp~-exp\_name~family\_small -device cuda \\
python3~-m~main.train~-data\_root~data/family\_small~-rule\_filename~rules\_exp\_60\%\
    _wrong.txt~-slice\_dim~16~-batchsize~16~-embedding\_size~256~-gcn\_free\_size~255~-
    load\_method~1~-exp\_folder~exp~-exp\_name~family\_small -device cuda \\
python3~-m~main.train~-data\_root~data/family\_small~-rule\_filename~rules\_exp\_80\%\
    _wrong.txt~-slice\_dim~16~-batchsize~16~-embedding\_size~256~-gcn\_free\_size~255~-
    load\_method~1~-exp\_folder~exp~-exp\_name~family\_small -device cuda \\
```

The hyperparameters have been taken from Github repository of ExpressGNN mentioned above. We took the same set of parameters for FB15k237 and the ones from Kinship to test all family datasets. ExpressGNN contains the following hyperparameters that are specified in the commands:

- slice_dim: slice dimension of posterior params

- batchsize: batch size for training

- gcn_free_size: embedding size of GCN concat param

- patience: patience for early stopping

- lr_decay_patience: learning rate decay factor

- entropy_temp: temperature for entropy term

## C.6. Uniker

The following commands can be used to run the experiments for Uniker. Note that the MLN_rule.txt file has to be changed with the new rules before running again the command for another rule set.

```
python3~run.py~family\_small~-1~family\_small\_model~TransE~8~0.0~0.2 \\
python3~run.py~family\_medium~-1~family\_medium\_model~TransE~8~0.0~0.2 \\
python3~run.py~wn18~-1~wn18\_model~TransE~8~0.0~0.2 \\
python3~run.py~fb15k~-1~fb15k\_model~TransE~8~0.0~0.2 \\
```

Here are the commands to test different levels of provided knowledge:

```
python3~run.py~family\_small~-1~family\_small\_model~TransE~8~0.0~0.2 \\
python3~run.py~family\_small~-1~family\_small\_model~TransE~8~0.0~0.2 \\
python3~run.py~family\_small~-1~family\_small\_model~TransE~8~0.0~0.2 \\
python3~run.py~family\_small~-1~family\_small\_model~TransE~8~0.0~0.2 \\
```

Here are the commands to test the robustness of the method against wrong knowledge:

```
python3~run.py~family\_small~-1~family\_small\_model~TransE~8~0.0~0.2 \\
python3~run.py~family\_small~-1~family\_small\_model~TransE~8~0.0~0.2 \\
python3~run.py~family\_small~-1~family\_small\_model~TransE~8~0.0~0.2 \\
python3~run.py~family\_small~-1~family\_small\_model~TransE~8~0.0~0.2 \\
```

The hyperparameters provided in the Github repository of Uniker are used, namely INTER=8, NOISE_THRESHOLD=0.0 (no noise), TOP_K_THRESHOLD=0.2 (threshold to include useful hidden triples).

### C.7. AnyBURL+RL

The following commands can be used to run the experiments with Anyburl+RL.

```
java -Xmx12G -cp AnyBURL-23-1.jar de.unima.ki.anyburl.Learn "config-learn$\_$family$\
    _$small.properties" \\
java -Xmx12G -cp AnyBURL-23-1.jar de.unima.ki.anyburl.Learn "config-learn$\_$family$\
    _$medium.properties" \\
java -Xmx12G -cp AnyBURL-23-1.jar de.unima.ki.anyburl.Learn "config-learn$\_$fb15k-237.
    properties" \\
java -Xmx12G -cp AnyBURL-23-1.jar de.unima.ki.anyburl.Learn "config-learn$\_$wn18.
    properties" \\
```

We used the following hyperparameters:

- PATH_TRAINING = data/family_small/train.txt

- PATH_OUTPUT = rules

- SNAPSHOTS_AT = 10, 50, 100    (time in seconds)

- WORKER_THREADS = 7

- POLICY = 2

- REWARD = 5

- EPSILON = 0.1

- THRESHOLD_CORRECT_PREDICTIONS = 2

- THRESHOLD_CONFIDENCE = 0.0001

- ZERO_RULES_ACTIVE = false    (rules that try to capture very simple relation-specific frequencies)

## Appendix D. Ontologies

### D.1. Family ontology

$wife(x, y) \Leftarrow husband(y, x)$
$husband(x, y) \Leftarrow wife(y, x)$
$father(x, z) \Leftarrow husband(x, y) \wedge mother(y, z)$
$mother(x, z) \Leftarrow wife(x, y) \wedge father(y, z)$

$father(x, z) \Leftarrow husband(x, y) \wedge son(z, y)$
$mother(x, z) \Leftarrow wife(x, y) \wedge son(z, y)$
$father(x, z) \Leftarrow husband(x, y) \wedge daughter(z, y)$
$mother(x, z) \Leftarrow wife(x, y) \wedge daughter(z, y)$
$aunt(x, z) \Leftarrow wife(x, y) \wedge uncle(y, z)$
$uncle(x, z) \Leftarrow husband(x, y) \wedge aunt(y, z)$
$uncle(x, z) \Leftarrow husband(x, y) \wedge nephew(z, y)$
$aunt(x, z) \Leftarrow wife(x, y) \wedge niece(z, y)$
$uncle(x, z) \Leftarrow husband(x, y) \wedge niece(z, y)$
$aunt(x, z) \Leftarrow sister(x, y) \wedge mother(y, z)$
$aunt(x, z) \Leftarrow sister(x, y) \wedge father(y, z)$
$uncle(x, z) \Leftarrow brother(x, y) \wedge mother(y, z)$
$uncle(x, z) \Leftarrow brother(x, y) \wedge father(y, z)$
$son(x, z) \Leftarrow son(x, y) \wedge wife(y, z)$
$son(x, z) \Leftarrow son(x, y) \wedge husband(y, z)$
$daughter(x, z) \Leftarrow daughter(x, y) \wedge wife(y, z)$
$daughter(x, z) \Leftarrow daughter(x, y) \wedge husband(y, z)$
$son(x, z) \Leftarrow brother(x, y) \wedge son(y, z)$
$son(x, z) \Leftarrow brother(x, y) \wedge daughter(y, z)$
$daughter(x, z) \Leftarrow sister(x, y) \wedge son(y, z)$
$daughter(x, z) \Leftarrow sister(x, y) \wedge daughter(y, z)$
$nephew(x, z) \Leftarrow son(x, y) \wedge sister(y, z)$
$nephew(x, z) \Leftarrow son(x, y) \wedge brother(y, z)$
$niece(x, z) \Leftarrow daughter(x, y) \wedge sister(y, z)$
$niece(x, z) \Leftarrow daughter(x, y) \wedge brother(y, z)$
$sister(x, z) \Leftarrow sister(x, y) \wedge sister(y, z)$
$sister(x, z) \Leftarrow sister(x, y) \wedge sister(z, y)$
$sister(x, z) \Leftarrow sister(x, y) \wedge brother(y, z)$
$sister(x, z) \Leftarrow sister(x, y) \wedge brother(z, y)$
$brother(x, z) \Leftarrow brother(x, y) \wedge brother(y, z)$
$brother(x, z) \Leftarrow brother(x, y) \wedge brother(z, y)$
$brother(x, z) \Leftarrow brother(x, y) \wedge sister(y, z)$
$brother(x, z) \Leftarrow brother(x, y) \wedge sister(z, y)$
$sister(x, z) \Leftarrow daughter(x, y) \wedge father(y, z)$
$sister(x, z) \Leftarrow daughter(x, y) \wedge mother(y, z)$
$brother(x, z) \Leftarrow son(x, y) \wedge father(y, z)$
$brother(x, z) \Leftarrow son(x, y) \wedge mother(y, z)$

## D.2. WN18RR ontology

$\_also\_see(x, y) \Leftarrow \_also\_see(y, x)$
$\_synset\_domain\_topic\_of(x, z) \Leftarrow \_derivationally\_related\_form(x, y) \wedge \_synset\_domain\_topic_o f(y, z)$
$\_synset\_domain\_topic\_of(x, z) \Leftarrow \_hypernym(x, y) \wedge \_synset\_domain\_topic\_of(y, z)$
$\_synset\_domain\_topic\_of(x, z) \Leftarrow \_hypernym(y, x) \wedge \_synset\_domain\_topic\_of(y, z)$
$\_synset\_domain\_topic\_of(x, z) \Leftarrow \_instance\_hypernym(x, y) \wedge \_synset\_domain\_topic\_of(y, z)$
$\_has\_part(x, z) \Leftarrow \_has\_part(x, y) \wedge \_instance\_hypernym(z, y)$
$\_derivationally\_related\_form(x, z) \Leftarrow \_derivationally\_related\_form(x, y) \wedge \_verb\_group(z, y)$
$\_derivationally\_related\_form(x, z) \Leftarrow \_verb\_group(y, x) \wedge \_derivationally\_related\_form(y, z)$

**D.3. FB15k237 ontology**

This can be find in our github repository, it is composed of 507 rules.

