# OpenReview forum: "A Comparative Analysis of Neurosymbolic Methods for Link Prediction"
_nesyconf.org/NeSy/2025/Conference_Phase_2 — NeSy 2025 - Phase 2 Poster_

### Official Review · Reviewer_otWh · 2025-06-24
**Review of submission 6**

**Rating:** 5
**Confidence:** 4

**Review:**

This paper presents a structured review and experimental benchmarking of state-of-the-art methods for link prediction on knowledge graphs. The paper highlights the limitations of both symbolic and subsymbolic methods, where the former are typically based on logical rules and the latter are based on methods such as knowledge graph embedding and graph neural networks. Neurosymbolic approaches are described as a way to bridge this gap, integrating embedding-based learning with symbolic reasoning to balance logical expressiveness and efficient computation.

The field of KG completion is growing rapidly, and a key contribution claimed by the authors is to provide a comprehensive experimental benchmarking of these methods under identical conditions, using the same datasets and metrics, to allow for fair comparisons.

The evaluation systematically varies factors including the choice of datasets, the number and composition of logical rules, and the proportion of incorrect rules within rule sets.

Methods are evaluation across different criteria, namely (1) accuracy (using MRR and HITS metrics); (2) scalability (in terms of graph size and number of rules where appropriate measured by total execution times for training and testing); (3) Impact of provided knowledge level on accuracy and (4) robustness concerning incorrect knowledge.

The evaluation was conducted on four datasets: Family small, Family medium, WN18RR, and FB15K237, which vary in size and rule characteristics.


Strengths of the paper

(1) The paper's aim is highly relevant to the field, as link prediction on KGs is important for various applications. The growing field of neurosymbolic AI for link prediction often sees methods evaluated independently, making direct comparisons difficult. This paper addresses a significant need for a rigorous and comprehensive experimental benchmarking to understand the practical strengths and limitations of these diverse approaches.

(2) I appreciate the commitment to fairly assess methods under identical experimental conditions. All experiments were performed using the same hardware and preprocessing steps, ensuring meaningful comparisons in terms of precision and execution time.

(3)  The authors evaluated a broad spectrum of link prediction methods, categorizing them. Representative approaches from each category were selected for testing. The evaluation was conducted on four diverse datasets.


Weaknesses:

(1) The paper states that methods are trained on the training set and evaluated on the test set, with statistics provided for train, test, and validation splits. However, it does not explicitly detail the methodology for splitting the KG into training and test data. In particular, splitting a KG into training and test data at random based on given proportions does not make any sense for approaches based on rules, because rules are inherently causal.

(2) The paper misses discussion of other relevant benchmarking approaches to KG completion such as the one by Liu et al. (KR 2023: 461-471). A more comprehensive review of existing benchmarking methodologies would strengthen the paper's contribution by positioning its approach within the broader landscape of evaluation strategies for KG completion.

(3) The use of HITS measures for "accuracy" is questionable and confusing, given that accuracy has a very precise definition in ML based on precision and recall. Also, the definition of HITS used in the paper should be made explicit. Also, HITS does not account for multiple correct answers, there is no penalty for incorrect high rank predictions and it is binary in nature. I would have expected in an evaluation/benchmarking paper a more in-depth discussion on various metrics and their appropriateness.

**Anonymity:**

Remain anonymous

---

### Official Review · Reviewer_8vnz · 2025-07-04

**Rating:** 5
**Confidence:** 4

**Review:**

This is a benchmarking paper exploring symbolic methods, subsymbolic methods and neural-symbolic methods for knowledge graph link prediction. 9 methods are compared with in total. It explores several aspects of these methods, including the accuracy, the scalability and the robustness to incorrect knowledge, with three datasets -- Family, WordNet, and FB15K237. This research topic is quite interesting, failing into the scope of the NeSy conference.

There are some weaknesses in this paper.

1. Regarding writing of the introduction, it provides no enough information of the content of this paper. E.g., it does not introduce what methods are evaluated, and what datasets are used. Some important observations or conclusions could also be introduced in the introduction.

2. The three adopted datasets have been widely explored, and are the ones where different knowledge graph link prediction methods are developed. It would be more interesting to see the performance on a different, larger-scale dataset, especially for checking the generalisation and real-world applicability of these methods.

3. There are many other more knowledge graph link prediction methods, especially the embedding and GNN-based methods. The selected methods (TransE, RotatE, BoxE and CompGCN) for evaluation may not be representative.

4. Some settings in the evaluation experiments may lead to biased conclusions. Symbolic methods rely on rules but the rules are mined by AMIE+. What if we use rules by other methods or sources? We can also set the hyper parameters to get different rules from AMIE+, leading to different performance for the symbolic method. Using the same computation for all the methods could also be unfair, as some methods rely more on GPU while some just use CPU.

5. The technical novelty of this paper is limited.

6. In 3.1, the title "Wordnet" should be changed to "WN18RR", aligned with "FB15k237".

**Anonymity:**

Remain anonymous

---

### Official Review · Reviewer_tkoy · 2025-07-08
**High-quality analysis paper; would benefit from presentation enhancements**

**Rating:** 7
**Confidence:** 3

**Review:**

This paper provides a taxonomy of link prediction methods organized into symbolic, subsymbolic, and neurosymbolic categories. Besides the taxonomy, the paper coins four experimental questions which are used to draw comparisons about the accuracy, scalability, and robustness of methods across the three families.

Strengths: The paper's topic is clearly of interest to NeSy. The taxonomy is intuitive and the additional levels for NeSy methods are particularly insightful. The experiments are informative and show the nuances in the landscape of methods and their interplay with dataset properties.

Weaknesses:
- I find the comparison to related work to be missing. The last paragraph of section 1 refers in passing to other surveys (presumably also focusing on link prediction methods?), stating that they lack experimental comparisons. However: 1) is it novel to perform such systematic comparisons of link prediction methods? and 2) it is unclear how novel the devised taxonomy is. Providing a paragraph with a comparison to prior work in section 1 or 2 would enrich the paper, in my opinion.
- Section 2 would benefit from a summary of the methods in a quick paragraph or a table at the end of the section, to capture some of their key properties described throuhgout the section. Similarly, Table 1 could benefit from incorporating some of the key qualitative dataset distinctions described in page 6 (and reflected again later in the Conclusion).

Minor:
- Please state why you opt for a transductive setting in 3.2.
- While FB15k and WN18RR are indeed larger than the Family sets, a key limitation is the exclusion of even larger and more heterogeneous datasets like Wikidata50M. It'd be good to briefly discuss this in section 5.

**Anonymity:**

Remain anonymous